# ATTENTION ENABLES ZERO APPROXIMATION ERROR

## ABSTRACT

Attention-based architectures become the core backbone of many state-of-the-art models for various tasks, including language translation and image classification. However, theoretical properties of attention-based models are seldom considered. In this work, we show that with suitable adaptations, the single-head self-attention transformer with a fixed number of transformer encoder blocks and free parameters is able to generate any desired polynomial of the input with no error. The number of transformer encoder blocks is the same as the degree of the target polynomial. Even more exciting, we find that these transformer encoder blocks in this model do not need to be trained. As a direct consequence, we show that the single-head self-attention transformer with increasing numbers of free parameters is universal. Also, we show that our proposed model can avoid the classical trade-off between approximation error and sample error in the mean squared error analysis for the regression task if the target function is a polynomial. We conduct various experiments and ablation studies to verify our theoretical results.

## 1 INTRODUCTION

By imitating the structure of brain neurons, deep learning models have replaced traditional statistical models in almost every aspect of applications, becoming the most widely used machine learning tools LeCun et al. (2015); Goodfellow et al. (2016). Structures of deep learning are also constantly evolving from fully connected networks to many variants such as convolutional networks Krizhevsky et al. (2012), recurrent networks Mikolov et al. (2010) and the attention-based transformer model Dosovitskiy et al. (2020). Attention-based architectures were first introduced in the areas of natural language processing, and neural machine translation Bahdanau et al. (2014); Vaswani et al. (2017); Ott et al. (2018), and now an attention-based transformer model has also become state-of-the-art in image classification Dosovitskiy et al. (2020). However, compared with significant achievements and developments in practical applications, the theoretical properties of attention-based transformer models are not well understood.

Let us describe some current theoretical progress of attention-based architectures briefly. The universality of a sequence-to-sequence transformer model is first established in Yun et al. (2019). After that, a sparse attention mechanism, BIGBIRD, is proposed by Zaheer et al. (2020) and the authors further show that the proposed transformer model is universal if its attention structure contains the star graph. Later, Yun et al. (2020) provides a unified framework to analyze sparse transformer models. Recently, Shi et al. (2021) studies the significance of different positions in the attention matrix during pre-training and shows that diagonal elements in the attention map are the least important compared with other attention positions. From a statistical machine learning point of view, the authors in Gurevych et al. (2021) propose a classifier based on a transformer model and show that this classifier can circumvent the curse of dimensionality.

The models considered in the above works all contain attention-based transformer encoder blocks. It is worth noting that the biggest difference between a transformer encoder block and a traditional neural network layer is that it introduces an inner product operation, which not only makes its actual performance better but also provides more room for theoretical derivations.

In this paper, we consider the theoretical properties of the single-head self-attention transformer with suitable adaptations. Different from segmenting $x$ into small pieces Dosovitskiy et al. (2020) and capturing local information, we consider a global pre-processing of $x$ and propose a new vector structure of the inputs of transformer encoder blocks. In this structure, in addition to the global

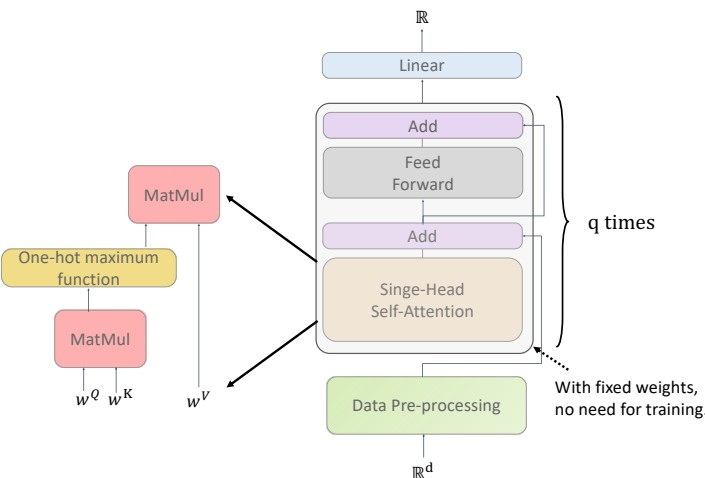

Figure 1: The Architecture of the single-head self-attention transformer. $W^Q, W^K, W^V$ stand for the query matrix, the key matrix, and the value matrix respectively. MatMul stands for the matrix multiplication.

information we obtain from data pre-processing, we place a one-hot vector to represent different features through the idea of positional encoding and place a zero vector to store the output values after each transformer encoder block. With such a special design, we can fix all transformer encoder blocks such that no training is needed for them. And it is able to realize the multiplication operation and store values in zero positions. By applying a well-known result in approximation theory Zhou (2018) stating that any polynomial $Q \in \mathcal{P}_q\left(\mathbb{R}^d\right)$ of degree at most $q$ can be represented by a linear combination of different powers of ridge forms $\xi_k \cdot x$ of $x \in \mathbb{R}^d$, we prove that the proposed model can generate any polynomial of degree $q$ with $q$ transformer encoder blocks and a fixed number of free parameters. As a direct consequence, we show that the proposed model is universal if we let the the number of free parameters and transformer encoder blocks go to infinity. Our theoretical results are also verified by experiments on synthetic data. In summary, the contributions of our work are as follows:

- We propose a new pre-processing method that captures global information and a new structure of input vectors of transformer encoder blocks.

- With the special structure of input of transformer encoder blocks, we can artificially design all the transformer encoder blocks in a spare way and prove that the single-head self-attention transformer with $q$ transformer encoder blocks and a fixed number of free parameters is able to generate any desired polynomial of degree $q$ of the input with no error.

- As a direct consequence, we show that the single-head self-attention transformer with increasing numbers of free parameters and transformer encoder block is universal.

- We conduct mean squared error analysis for the regression task with our proposed model. We show that if the target function is a polynomial, our proposed model can avoid the classical trade-off between approximation error and sample error. And the convergence rate is only controlled by the number of samples if we treat $d$ and $q$ as constants.

- We apply our model to noisy regression tasks with synthetic data and real-world data-set. Our experiments show that the proposed model performs much better than traditional fully connected neural networks with a comparable number of free parameters.

- We apply our model to the image classification task and achieve better performance than Vision Transformer on the CIFAR-10 data set with suitable adaptations.

## 2 TRANSFORMER STRUCTURES

In this section, we formally introduce the single-head self-attention transformer considered in this paper. The overall architecture is shown in Figure 1.

## 2.1 Data pre-processing

For an input $x \in \mathbb{R}^d$ which can be a vector or the concatenation of an image, the usual pre-processing method is to segment it into small pieces and then conduct linear transforms, which can be thought of as extracting local features. However, we propose to directly apply a full matrix $F \in \mathbb{R}^{n \times d}$ to get global features $Fx = t \in \mathbb{R}^n$, where $F = [\xi_1, \cdots, \xi_n]^\top$ with $\xi_i \in \mathbb{R}^d$ and $\|\xi_i\| \leq 1$. The matrix $F$ is obtained through the training process. Then we have $n$ global features $t_i = \langle \xi_i, x \rangle$ of the input $x$. Now we introduce the structure of inputs for transformer encoder blocks as follows,

$$z_i = [t_i, \overbrace{0, \cdots, 0, \underbrace{1}_{(i+1)-\text{th entry}}, 0, \cdots, 0}^{n}, \overbrace{0, \cdots, 0}^{q}, 1]^\top,$$

for $i = 1, \cdots, n$. Each one of them is a sparse vector in $\mathbb{R}^{n+q+2}$ and all the $n$ vectors are inputs for the transformer encoder blocks. As we have covered before, we put a one-hot vector of dimension $n$ inside $z_i$ representing different features $t_i$ of the input $x$, which is similar to the idea of positional encoding. And we also place a $q$ dimensional zero vector to store outputs from each transformer encoder block. At the last position, we place a constant 1 for the ease of computation in transformer encoder blocks. We use $\mathcal{F}(x) : \mathbb{R}^d \to \mathbb{R}^{(n+q+2) \times (n)}$ to denote the above transformation such that $\mathcal{F}(x) = [z_1, \cdots, z_n]$.

## 2.2 Single-dead self-attention transformer encoder blocks

One transformer encoder block contains a self-attention layer and a fully connected layer with a linear transformation. In the self-attention layer, we have one query matrix $W^Q \in \mathbb{R}^{(n+1) \times (n+q+2)}$, one key matrix $W^K \in \mathbb{R}^{(n+1) \times (n+q+2)}$, and one value matrix $W^V \in \mathbb{R}^{(n+q+2) \times (n+q+2)}$.

For every input $z_i$, we calculate the query vector $q_i = W^Q z_i \in \mathbb{R}^{n+1}$, the key vector $k_i = W^K z_i \in \mathbb{R}^{n+1}$, and the value vector $v_i = W^V z_i \in \mathbb{R}^{n+q+2}$. With all these values, we have $n$ attention vectors

$$\alpha_i = [\langle q_i, k_1 \rangle, \cdots, \langle q_i, k_i \rangle, \cdots, \langle q_i, k_n \rangle]^\top \in \mathbb{R}^n.$$

In our proposed model, the soft-max function in the self-attention layer is replaced by a one hot maximum function $\hat{m}(\alpha_i) : \mathbb{R}^n \to \mathbb{R}^n$ which keeps the largest value unchanged and sets the other values to 0. We use the notation $\mathcal{A}_{W^Q, W^K, W^V} : \mathbb{R}^{(n+q+2) \times n} \to \mathbb{R}^{(n+q+2) \times n}$ to denote the mapping of the self-attention layer. Then the output of the self-attention layer is given by $\mathcal{A}_{W^Q, W^K, W^V}(z_1, \cdots, z_n) = [\hat{z}_1, \cdots, \hat{z}_n]$, where $\hat{z}_i = z_i + W^V Z \hat{m}(\alpha_i)$, with $Z = [z_1, \cdots, z_n]$.

The fully connected layer with a linear transformation contains two matrices $W_1 \in \mathbb{R}^{2 \times (n+q+2)}$, and $W_2 \in \mathbb{R}^{(n+q+2) \times 2}$, and two bias vectors $b_1 \in \mathbb{R}^2, b_2 \in \mathbb{R}^{n+q+2}$. We use the notation $\mathcal{B}_{W_1, W_2, b_1, b_2} : \mathbb{R}^{(n+q+2) \times n} \to \mathbb{R}^{(n+q+2) \times n}$ to denote the mapping of the fully connected layer with a linear transformation. Then we have $\mathcal{B}_{W_1, W_2, b_1, b_2}(\hat{z}_1, \cdots, \hat{z}_n) = [z'_1, \cdots, z'_n]$, where $z'_i = \hat{z}_i + W_2 \sigma(W_1 \hat{z}_1 + b_1) + b_2$, and $\sigma$ is the ReLU activation function acting component-wise.

Now we define our single-head self-attention transformer model with $\ell$ transformer encoder blocks as

$$\mathcal{T}^\ell(x) = \mathcal{B}^\ell \circ \mathcal{A}^\ell \circ \cdots \circ \mathcal{B}^1 \circ \mathcal{A}^1 \circ \mathcal{F}(x),$$

where $\mathcal{F}, \mathcal{A}^i, \mathcal{B}^i$ are the mappings defined above. We further concatenate the output matrix into one vector and apply a linear transformation with a bias term to get our final output, that is,

$$\mathcal{C}^\ell(x) = \beta \cdot \textbf{concat}\left(\mathcal{T}^\ell(x)\right) + b,$$

with $\beta \in \mathbb{R}^{n(n+q+2)}$ and $b \in \mathbb{R}$. We require the vector $\beta$ to possess a sparse structure which will be shown in the proof. The values in $\beta$ and $b$ are obtained through the training process. The layer normalization is not considered in our model.

# 3 Main results

## 3.1 Zero approximation error

In this section, we first present our main result from an approximation theory point of view, showing that the single-head self-attention transformer model can generate any desired polynomial with a fixed number of transformer encoder blocks and free parameters. Before stating our main theorem,

we first present two important lemmas. For the following lemma, we construct a sparse single-head self-attention block with a fixed design that is able to realize the multiplication operation and store different products in the output vectors simultaneously.

**Lemma 3.1.** *For all $n$ input vectors in the form of*

$$z_i = [t_i, e_i, \overbrace{x_i, y_i, 0, \cdots, 0}^{q}, 1]^\top \in \mathbb{R}^{(n+q+2)\times 1},$$

*with $t_i, x_i, y_i \in \mathbb{R}$ and absolute values bounded by some known constant $M$ for $i = 1, \cdots, n$, there exists a sparse single-head self-attention transformer encoder block with fixed matrices $W^Q$, $W^K$, $W^V$, $W_1$, $W_2$ and vectors $b_1$, $b_2$ that can produce output vectors as*

$$z_i' = [t_i, e_i, \overbrace{x_i, y_i, -x_i y_i, 0, \cdots, 0}^{q}, 1]^\top \in \mathbb{R}^{(n+q+2)\times 1},$$

*where $e_i$ denotes the one-hot vector of dimension $n$ with value $1$ in the $i$-th position of $e_i$. The softmax function is replaced by one hot maximum function. The number of non-zero entries is $2n + 8$.*

**Remark 3.2.** *The above lemma shows that a fixed single-head self-attention transformer encoder block is able to simultaneously calculate the product of two elements in all $n$ input vectors within the same two entries and store the negative value in the same $0$ positions. Since the construction is fixed, these transformer encoder blocks in the whole model do not need to be trained.*

Now we introduce a well-known result in approximation theory showing that any polynomial function $Q \in \mathcal{P}_q(\mathbb{R}^d)$ of degree at most $q$ can be represented by a linear combination of different powers of ridge forms $\xi_k \cdot x$ of $x \in \mathbb{R}^d$. The following lemma is first presented and proved in Zhou (2018) and also plays an important role in the analysis of deep convolutional neural networks Zhou (2020); Mao et al. (2021).

**Lemma 3.3.** *Let $d \in \mathbb{N}$ and $q \in \mathbb{N}$. Then there exists a set $\{\xi_k\}_{k=1}^{n_q} \subset \{\xi \in \mathbb{R}^d : \|\xi\| = 1\}$ of vectors with $\ell_2-norm$ $1$ such that for any $Q \in \mathcal{P}_q(\mathbb{R}^d)$ we can find a set of coefficients $\{\beta_{k,s} : k = 1, \cdots, n_q, s = 1, \cdots, q\} \subset \mathbb{R}$ such that*

$$Q(x) = Q(0) + \sum_{k=1}^{n_q} \sum_{s=1}^{q} \beta_{k,s} (\xi_k \cdot x)^s, \quad x \in \mathbb{R}^d, \tag{1}$$

*where $n_q = \binom{d-1+q}{q}$ is the dimension of $\mathcal{P}_q^h(\mathbb{R}^d)$, the space of homogeneous polynomials on $\mathbb{R}^d$ of degree $q$.*

**Remark 3.4.** *The above lemma shows that any polynomial $Q \in \mathcal{P}_q(\mathbb{R}^d)$ can be uniquely determined by $Q(0)$, $\beta_{k,s}$ and $\xi_k$. So by applying the above lemma, we can perfectly reproduce any polynomial with proper construction.*

Now we are ready to state our main result on the single-head self-attention transform model.

**Theorem 3.5.** *Let $B > 0$ and $q \in \mathbb{N}$. For any polynomial function $Q \in \mathcal{P}_q(\mathbb{R}^d)$ of degree at most $q$, there exist a single-head self-attention transformer model with $q$ transformer encoder blocks such that the output function $\mathcal{C}^q$ equals $Q$ on $\{x \in \mathbb{R}^d : \|x\| \leq B\}$*

$$\mathcal{C}^q(x) = Q(x), \ \forall \|x\| \leq B.$$

*The number of free parameters is less then $d^{q+1} + qd^q + 1$ which comes from $F$, $\beta$ and $b$. The number of non-zero entries in this model is less than $d^{q+1} + 3qd^q + 8q + 1$.*

**Remark 3.6.** *The above theorem shows a very strong property of the self-attention transformer model that it can generate any desired polynomial with a finite number of free parameters. As we can see, the degree of the polynomial is reflected in the number of transformer encoder blocks, showing that the more blocks the transformer have, the more complex polynomial it can represent. Clearly, this result outperforms that of the other classical deep learning models without attention-based structure in at least two aspects. First, since the linear combination of the output units of traditional ReLU neural networks is only a piece-wise linear function of the input, no matter how many finite layers and free parameters, it can never produce a polynomial of the input with no error. Second, the transformer encoder blocks in our construction only serve as the realization of the multiplication operation. The non-zero values are all pre-designed constants, so no training is needed for these blocks. We only need to train free parameters in $F$, $\beta$, and $b$.*

As a direct consequence of the above result, the proposed single-head self-attention transform model is universal.

**Corollary 3.7.** *Let $d \in \mathbb{N}$ and $q \in \mathbb{N}$. For any bounded continuous function $f$ on $[0,1]^d$, there exists a single-head self-attention transformer with increasing numbers of free parameters and transformer encoder blocks such that*

$$\lim_{q \to \infty} \|\mathcal{C}^q - f\|_{C([0,1]^d)} = 0$$

The above result is a simple application of the denseness of the polynomial set, which shows that the transformer model discussed in our paper is universal if we let the number of free parameters and transformer encoder blocks go to infinity.

### 3.2 MEAN SQUARED ERROR ANALYSIS

In this subsection, we consider the regression problem with mean squared error in the setting of statistical learning theory with our single-head self-attention transformer model. Let $\mathcal{X} := \{x : \|x\|_2 \leq B, x \in \mathbb{R}^d\}$ and $\mathcal{Y} \subset \mathbb{R}$. We observe $N$ i.i.d. vectors $x_i \in \mathcal{X}$ with an unknown probability distribution $\rho_X$ and $N$ responses $y_i \in \mathbb{R}$ from the model

$$y_i = f_\rho(x_i) + \epsilon_i, \quad i = 1, \cdots, N,$$

where the noise variables $\epsilon_i$ are assumed to satisfy $\mathbb{E}(\epsilon_i | x_i) = 0$. Usually the noise variables satisfy standard normal distribution. We denote the joint distribution of $(x_i, y_i)$ by $\rho$ and $D = \{(x_i, y_i)\}_{i=1}^N$ is drawn from the probability measure $\rho$. Our goal is to recover the regression function $f_\rho$ from the sample $D$. For any measurable function, we can define the population risk as $\mathcal{E}(f) := \int_{\mathcal{X} \times \mathcal{Y}} (f(x) - y)^2 d\rho$, and the empirical risk as $\mathcal{E}_D(f) := \frac{1}{N} \sum_{i=1}^N (f(x_i) - y_i)^2$. We use $f_{D,\mathcal{H}} = \arg\min_{f \in \mathcal{H}} \mathcal{E}_D(f)$, to approximate $f_\rho$ where $\mathcal{H}$ is the hypothesis space generated by our proposed model given by

$$\mathcal{H} := \mathcal{H}_{\ell, \tilde{B}} := \left\{ \mathcal{C}^\ell(x) : \|\xi_i\| \leq 1, \|\beta\|_\infty \leq \tilde{B}, \|b\|_\infty \leq \tilde{B} \right\}. \tag{2}$$

Let $\left( L^2_{\rho_X}, \|\cdot\|_{\rho_X} \right)$ be the space of $\rho_X$ square-integrable functions on $\mathcal{X}$ such that $\|f\|^2_{\rho_X} := \int_{\mathcal{X}} f^2(x) d\rho_X$. The target of mean squared error analysis is to derive convergence rate of $\mathcal{E}(f_{D,\mathcal{H}}) - \mathcal{E}(f_\rho) = \|f_{D,\mathcal{H}} - f_\rho\|^2_{\rho_X}$. Now we are ready to state our main result of mean squared error analysis.

**Theorem 3.8.** *Let $B, \tilde{B} > 0$. Let $d \in \mathbb{N}^+$, $q \in \mathbb{N}$, $\mathcal{H}$ be defined as (2) and $f_\rho$ be a polynomial of degree at most $q$ on $\mathcal{X}$ such that $|f_\rho| \leq 2qn_q\tilde{B}B^q$. Then for $N \in \mathbb{N}$ such that $N \geq n_q$ we have*

$$\mathbb{E} \|f_{D,\mathcal{H}} - f_\rho\|^2_{\rho_X} \leq \tilde{C}_{q,d,\tilde{B},B} \frac{n_q^3 \log N}{N},$$

*where $\tilde{C}_{q,d,\tilde{B},B}$ is a constant depending on $q, d, \tilde{B}, B$ and $n_q = \binom{d-1+q}{q}$ is the dimension of $\mathcal{P}_q^h(\mathbb{R}^d)$.*

**Remark 3.9.** *For classical mean squared error analysis of the regression task, there is a trade-off between approximation error and sample error and the convergence rate is controlled by the maximum value of these two errors. Since our model can achieve zero approximation error for polynomial target functions, our main result shows that this trade-off can be avoided. As a direct consequence, the convergence rate $\mathcal{O}\left(\frac{\log N}{N}\right)$ is only controlled by the number of samples if we treat $d$ and $q$ as constants. The above result is also verified in our experiment 5.2.*

## 4 COMPARISON AND DISCUSSION

In this section, we compare our work with some existing theoretical results on the transformer model Yun et al. (2019; 2020); Zaheer et al. (2020); Shi et al. (2021). Since these works use similar methods to those in Yun et al. (2019), we focus on the theoretical contributions of this paper.

In Yun et al. (2019), the authors show that transformer models are universal approximators of continuous sequence-to-sequence functions with compact support with trainable positional encoding.

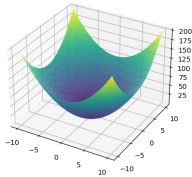 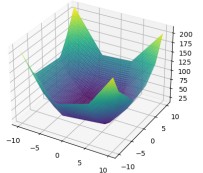 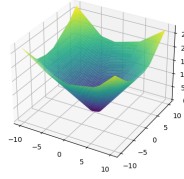

(a) ATTENTION with MSE 0.02. (b) $\text{NN}_{depth}$ with MSE 134.23. (c) $\text{NN}_{width}$ with MSE 10237.27.

Figure 2: For the target polynomial $f_{\rho,1}$, the above 3-D surface plots are output functions of three different models after the training process. ATTENTION stands for our single-head self-attention transformer model, while $\text{NN}_{depth}$ and $\text{NN}_{width}$ stand for fully connected neural networks illustrated in experimental setting 5.1. MSE stands for the Mean Squared Error evaluated at testing data.

The notion of contextual mappings is also formalized, and it is shown that the attention layers can compute contextual mappings, where each unique context is mapped to a unique vector.

The universality result is achieved in three key steps: **Step 1**. Approximate continuous permutation equivariant functions with piece-wise constant functions $\bar{\mathcal{F}}_{PE}(\delta)$. **Step 2**. Approximate $\bar{\mathcal{F}}_{PE}(\delta)$ with modified Transformers $\bar{\mathcal{T}}$. **Step 3**. Approximate modified Transformers $\bar{\mathcal{T}}$ with original Transformers $\mathcal{T}$.

In order to express the above steps more clearly, we show the idea of proof as follows. For an input $X \in \mathbb{R}^{d \times n}$, the authors first use a series of feed-forward layers that can quantize $X$ to an element $L$ on the extended grid $\mathbb{G}_\delta^+ := \left\{ -\delta^{-nd}, 0, \delta, \cdots, 1-\delta \right\}^{d \times n}$. Activation functions that are applied to these layers are piece-wise linear functions with at most three pieces, and at least one piece is constant. Then, the authors use a series of self-attention layers in the modified transformer network to implement a contextual mapping $q(L)$. After that, a series of feed-forward layers in the modified transformer network can map elements of the contextual embedding $q(L)$ to create a desired approximator $\bar{g}$ of the piece-wise constant function $\bar{f} \in \bar{\mathcal{F}}_{PE}(\delta)$ which is the approximator of the target function.

We would like to address major differences between our work and theirs. First, the output functions are different. In the above work, the goal is to approximate a continuous function defined from $\mathbb{R}^{n \times d}$ to $\mathbb{R}^{n \times d}$, which focuses on sequence-to-sequence functions. In our setting, we use the linear combination of the units in the last layer as our output, which focuses on regression and classification tasks. Second, the two structures we consider are slightly different. The self-attention layers and feed-forward layers in their transformer model are set in an alternate manner. Although this may explain the different functions of different types of layers, it changes the structure of the transformer model in real applications. In our setting, we guarantee the integrity of transformer encoder blocks and analyze each transformer encoder block as a whole. Last but not least, the ultimate goals and core ideas of the theoretical analysis of our two papers are different. Because the inner product operation is the biggest difference between the attention layer and the traditional network layer, we focus on this special structure for analysis. We find that if we can make good use of this inner product structure, then from the perspective of theoretical analysis, we do not have to think about approximation but can directly generate the function we want. And the exact construction only requires a finite number of free parameters with fixed transformer encoder blocks. This shows the different thinking in our theory and distinguishes our method from using piece-wise functions to approximate target functions.

## 5 EXPERIMENTS

In this section, we first verify our main results of our single-head self-attention transformer model by experiments on two groups of synthetic data. Then, we conduct several ablation studies to demonstrate the superiority of the self-attention transformer.

## 5.1 EXPERIMENTS ON POLYNOMIAL FUNCTIONS AND REAL-WORLD DATA-SET

**Target functions** For these two experiments, we consider the noisy regression task

$$y = f_\rho(\mathbf{x}) + \epsilon,$$

where $f_\rho$ is the target polynomial and $\epsilon$ is the standard normal noise.

For the first experiment, in order to visualize the advantages of our proposed model, we consider a simple polynomial,

$$f_{\rho,1}(\mathbf{x}) = x_1^2 + x_2^2,$$

which satisfies $d = 2$ and $q = 2$.

For the second experiment, to show the strong expressiveness of our model, we consider a complicated polynomial

$$f_{\rho,2}(\mathbf{x}) = x_1^5 + 3x_2^4 + 2x_3^3 + 5x_3x_4 + 3x_5^2 + 2x_6x_7x_8 + 2x_9,$$

which satisfies $d = 10$ and $q = 5$.

**Experimental setting** To demonstrate the power of attention-based structures, we compare our proposed model with two types of ReLU fully connected neural networks with a comparable number of free parameters. Since for a polynomial $Q$ of degree $q$, our proposed model has one linear transformation with matrix $F \in \mathbb{R}^{n_q \times d}$ and $q$ transformer encoder blocks, we use $\text{NN}_{depth}$ to denote the fully connected network with $q + 1$ layers, and we use $\text{NN}_{width}$ to denote the shallow net with $n_q$ units in the hidden layer. For these two fully connected networks, we use the same way as our ATTENTION model to generate output value, which is the linear combination of units in the last layer with a bias term. The detailed descriptions of the experiments, i.e., data generating process, training hyper-parameter, and model architectures, can be found in B.

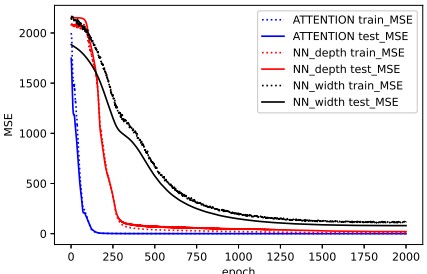

Figure 3: A comparison of the convergence speed and generalization gap between our single-head self-attention model and two types of fully connected neural networks.

Table 1: A comparison of three models learning $f_{\rho,2}$. $\text{MSE}_{Tr}$ and $\text{MSE}_{Te}$ stand for the mean-squared error of the training data and the testing data after 2000 epochs training, respectively. We say that a model achieves convergence if the absolute difference of $\text{MSE}_{Tr}$ of two consecutive epochs is less than 0.01. # EPOCHS stands for the number of epochs the model used before achieving convergence, and RUN TIME represents the corresponding running time of the training process.

|  | $\text{MSE}_{Tr}$ | $\text{MSE}_{Te}$ | # EPOCHS | RUN TIME[1] |
|---|---|---|---|---|
| ATTENTION | **0.938** | **0.109** | **212** | **1.9** |
| $\text{NN}_{depth}$ | 5.884 | 103.916 | 956 | 7.6 |
| $\text{NN}_{width}$ | 50.282 | 35.662 | 329 | 2.4 |

**Experimental results** For the target polynomial $f_{\rho,1}$, Figure 2 demonstrates the strong power of learning polynomials of our proposed model. With only 19 free parameters, our single-head self-attention transformer can perfectly capture the target function by using noisy data. Due to the nature of piece-wise linear output function, both two types of fully connected neural networks obviously can not achieve comparable results with very few parameters.

---

[1] GPU * min on NVIDIA A100 Tensor Core GPU.

For the target function $f_{\rho,2}$, Table 1 and Figure 3 also demonstrate the superior ability of our model to learn a complicated polynomial. Our single-head self-attention transformer is the only one that can fit the ground truth function exactly with good convergence speed. Moreover, our model has a much better generalization power than both two types of fully connected neural networks with a comparable number of free parameters.

## 5.2 THE SIGNIFICANCE OF THE SELF-ATTENTION TRANSFORMER

In this part, we conduct several experiments to address the following question:

- Given that neural networks can approximate any continuous function with enough width, what is the significance of zero approximation error achieved by the self-attention transformer?
- Can zero approximation error be achieved by using the quadratic activation function instead of the Relu or Sigmoid activation functions?
- Can our ATTENTION model avoid the trade-off between approximation error and sample error practically?
- Can our ATTENTION model achieve good performance on real-world data-set?

Table 2: The comparison of Mean Squared Error for learning $f_{\rho,1}$. Column ATTENTION demonstrates the MSE achieved by our architecture with 13 free parameters, while Column WIDTH=$10^1$ to Column WIDTH=$10^5$ demonstrates the MSE achieved by the fully connected neural networks with one hidden layer and corresponding width.

|  | ATTENTION | WIDTH=$10^1$ | WIDTH=$10^2$ | WIDTH=$10^3$ | WIDTH=$10^4$ | WIDTH=$10^5$ |
|---|---|---|---|---|---|---|
| MSE$_{\text{TRAIN}}$ | **1.11** | 7796.18 | 285.03 | 21.34 | 141.81 | 1048.91 |
| MSE$_{\text{TEST}}$ | **0.02** | 6713.48 | 333.21 | 49.44 | 73.21 | 1099.98 |

**The significance of zero approximation error**  Table 2 demonstrates the huge gap between universal approximation and zero approximation error. Even in the simple two-dimensional setting for learning $f_{\rho,1} = x_1^2 + x_2^2$, a traditional neural network can not perfectly approximate $f_{\rho,1}$ given large numbers of width through training. The performance of the ultra-wide vanilla neural network with width=$10^5$ even drops dramatically.

Table 3: The comparison of Mean Squared Error for learning $f_{\rho,3} = (x_1 + x_2 + x_3 + x_4 + x_5)^5$. Column ATTENTION demonstrates the MSE achieved by our architecture, while Column NN$_{depth}$ and NN$_{width}$ demonstrates the MSE achieved by the fully connected neural networks with comparable amount of parameters.

|  | ATTENTION | NN$_{depth}$ | NN$_{width}$ |
|---|---|---|---|
| MSE$_{\text{TRAIN}}$ | **1.67** | 177282.67 | 2440869.38 |
| MSE$_{\text{TEST}}$ | **0.81** | 196548.68 | 2246920.0 |

Moreover, the gap between zero approximation error and universal approximation error can be large enough in the sense that the former can perfectly fit the target function while the latter can even diverge. This phenomenon can happen even for the $d = 5$ and $q = 5$ case, which is demonstrated in the Table 3.

Table 4: The comparison of Mean squared error for learning $f_{\rho,4} = x_1^3 + x_2^3$. Column ATTENTION demonstrates the MSE achieved by our architecture with 21 free parameters, while Column WIDTH=$10^3$ to Column WIDTH=$10^5$ demonstrates the MSE achieved by the fully connected neural networks with one hidden layer and corresponding width.

|  | ATTENTION | WIDTH=$10^2$ | WIDTH=$10^3$ | WIDTH=$10^4$ |
|---|---|---|---|---|
| MSE$_{\text{TRAIN}}$ | **0.99** | 13.29 | 12.97 | 12.41 |
| MSE$_{\text{TEST}}$ | **0.0002** | 11.57 | 11.56 | 11.29 |

**Zero approximation cannot be achieved by the quadratic activation function** Table 4 shows the performance of different kinds of fully connected networks with the quadratic activation function. It is clear that Fully Connected networks with the quadratic activation function can not achieve zero approximation error, which demonstrates the superiority of the self-attention transformer.

**The self-attention transformer can avoid the trade-off between the approximation error and the sample error** Figure 4 shows the mean squared error of our single-head self-attention model under a different number of samples. The test mean squared error monotone decreases as the number of samples increases. Also, the training error monotone increases from 0.1349 to 0.9660, which indicates our model prevents over-fitting the noise.

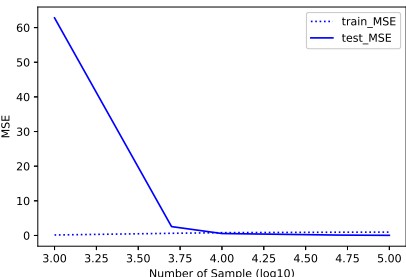

Figure 4: A demonstration of the test error of our single-head self-attention model under different number of samples. The x-axis is the base 10 logarithm of the number of training samples.

**Performance on real-world data-set and beyond regression task** Theorem 3.5 and Theorem 3.8 show that our proposed model can fit any desired polynomial function with no error. The aforementioned experiments verify our theorems on fitting polynomial functions. Two questions then arise naturally. 1) Whether our proposed ATTENTION model works well on real-world data-set? 2) Whether our proposed ATTENTION model works well beyond regression task, for example, on image classification task? We first conduct experiments on Household Electric Power Consumption Data-set on Appendix B.3 and achieve better performance than fully connected neural networks with a comparable number of free parameters. Moreover, we apply our model to the image classification task and achieve better performance than Vision Transformer on the CIFAR-10 data set with suitable adaptations. The detailed description can be found on Appendix B.4.

## 6 CONCLUSION

In this paper, we introduced a single-head self-attention transformer model and showed that any polynomial could be generated exactly by an output function of such a model with the number of transformer encoder blocks equal to the degree of the polynomial. The transformer encoder blocks in this model do not need to be trained. We also show that if the target is a polynomial function, our proposed model can avoid the classical trade-off between approximation error and sample error for the regression task.

In the future, many research directions will be very attractive. First of all, our core idea is different from the traditional one of approximation, and through the appropriate adjustment of the transformer model, a completely new theoretical result is presented. Also, in our structure, the transformer encoder blocks are completely fixed. It is of great interest to check our results in real applications to see whether these adaptations can indeed bring benefits. Second, we have obtained such exciting theoretical results by considering only the single-head self-attention structure. We can consider whether the multi-head structure can lead to more surprising conclusions. Last but not least, it is of great interest to consider this model under the setting of statistical machine learning. As we can see in our experiments, as long as the number of free parameters meets the theoretical requirement, our model can not only learn the objective function well but also has a much stronger generalization ability than other models. And as far as we are concerned, this is the first deep learning model which is capable of reaching zero approximation error for certain function classes.

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

## A APPENDIX: PROOF OF MAIN RESULTS

*Proof of Lemma 3.1.* We present explicit constructions of matrices and biases in single-head self-attention transformer encoder block. We let $W^Q \in \mathbb{R}^{(1+n) \times (2+n+q)}$ as follows,

$$
W^Q = \begin{bmatrix} 0 & \cdots & 0 & 1 & 0 & \cdots & 0 & 0 \\ 0 & 2M^2 & 0 & 0 & \cdots & \cdots & 0 & 0 \\ 0 & 0 & 2M^2 & 0 & \ddots & \ddots & 0 & 0 \\ 0 & 0 & 0 & \ddots & \ddots & \cdots & 0 & 0 \\ 0 & 0 & 0 & 0 & 2M^2 & 0 & \cdots & 0 \end{bmatrix},
$$

where the constant 1 in the first row is in the $(n+2)-$th column. And we set $W^Q_{(t,t)} = 2M^2$ for $t = 2, \cdots, n+1$ and all the other elements 0. Since the inputs are in the form of

$$
z_i = [t_i, e_i, \overbrace{x_i, y_i, 0, \cdots, 0}^{q}, 1]^\top \in \mathbb{R}^{(n+q+2) \times 1},
$$

where $e_i$ denotes the one-hot vector of dimension $n$ with value 1 in the $i$-th position of $e_i$. Then we know that $q_i \in \mathbb{R}^{(1+n) \times 1}$ is as follows

$$
q_i = W^Q z_i = [x_i, \underbrace{0, \cdots, 0, \overbrace{2M^2}^{(i+1)\text{-th entry}}, 0, \cdots, 0}_{n}]^\top.
$$

We let $W^K \in \mathbb{R}^{(1+n) \times (2+n+q)}$ as follows

$$
W^K = \begin{bmatrix} 0 & \cdots & 0 & 1 & \cdots & \cdots & 0 & 0 \\ 0 & 1 & 0 & 0 & \cdots & \cdots & 0 & 0 \\ 0 & 0 & 1 & 0 & \ddots & \ddots & 0 & \vdots \\ 0 & 0 & 0 & \ddots & \ddots & \cdots & 0 & 0 \\ 0 & 0 & 0 & 0 & 1 & 0 & \cdots & 0 \end{bmatrix}.
$$

where the constant 1 in the first row is in the $(n+3)-$th column. And we set $W^K_{(t,t)} = 1$ for $t = 2, \cdots, n+1$ and other elements 0. Then we have $k_i \in \mathbb{R}^{(1+n) \times 1}$ as

$$
k_i = W^Q z_i = [y_i, \underbrace{0, \cdots, 0, \overbrace{1}^{(i+1)\text{-th entry}}, 0, \cdots, 0}_{n}]^\top.
$$

We can easily find that for each $i$, if $j = i$, then $\langle q_i, k_j \rangle = x_i y_i + 2M^2$. And if $j \neq i$, then $\langle q_i, k_j \rangle = x_i y_j$. By the condition $|x_i| < M$ and $|y_i| < M$, clearly we have $x_i y_i + 2B^2 > x_i y_j$.

Then the attention vector $\alpha_i$ is

$$
\alpha_i = [\langle q_i, k_1 \rangle, \cdots, \langle q_i, k_i \rangle, \cdots, \langle q_i, k_n \rangle]^\top \in \mathbb{R}^{n \times 1}.
$$

Since we apply the one hot maximum function to $\alpha_1$, then by the construction we have

$$
\hat{\alpha}_i = [0, \cdots, 0, \overbrace{x_i y_i + 2B^2}^{i-\text{th entry}}, 0, \cdots, 0]^\top \in \mathbb{R}^{n \times 1}.
$$

For the matrix $W^V \in \mathbb{R}^{(n+q+2) \times (n+q+2)}$, we set

$$
W^V_{i,j} = \begin{cases} 1, & i = n+4, j = n+q+2, \\ 0, & \text{others.} \end{cases}
$$

Then we know that for $i = 1, \cdots, n$,

$$
W^V z_i = [0, \cdots, 0, \overbrace{1}^{(n+4)-\text{th entry}}, 0, \cdots, 0]^\top.
$$

By the equation
$$\hat{z}_i = z_i + W^V Z \hat{\alpha}_i,$$
we know that the outputs $z_i \in \mathbb{R}^{(n+q+2)\times 1}$ of self-attention layer are
$$\hat{z}_i = [t_i, e_i, \overbrace{x_i, y_i, x_i y_i + 2M^2, 0, \cdots, 0}^{q}, 1]^\top,$$
where $e_i$ denotes the one-hot vector of dimension $n$ with value 1 in the $i$-th position of $e_i$.

Now we construct the fully connected layer in the transformer. For $W_1 \in \mathbb{R}^{2\times(2+n+q)}$, we let
$$W_{1,(i,j)} = \begin{cases} 1, & i = 1, j = n+4, \\ -1, & i = 2, j = n+4, \\ 0, & \text{others.} \end{cases}$$
and $b_1 = [0,0]^\top$. Then we have
$$\sigma(W_1 z_i + b_1) = [\sigma(x_i y_i + 2M^2), \sigma(-x_i y_i - 2M^2)]^\top.$$
For $W_2 \in \mathbb{R}^{(n+q+2)\times 2}$, we let
$$W_{2,(i,j)} = \begin{cases} -2, & i = n+4, j = 1, \\ 2, & i = n+4, j = 2, \\ 0, & \text{others.} \end{cases}$$
And we let $b_2 \in \mathbb{R}^{(n+q+2)\times 1}$ to be
$$b_{2,(i)} = \begin{cases} 2M^2, & i = n+4, \\ 0, & \text{others.} \end{cases}$$
Then by
$$z_i' = \hat{z}_i + W_2 \sigma(W_1 \hat{z}_1 + b_1) + b_2,$$
we have
$$z_i' = [t_i, e_i, \overbrace{x_i, y_i, -x_i y_i, 0, \cdots, 0}^{q}, 1]^\top \in \mathbb{R}^{(n+q+2)\times 1}.$$
Since we assume that $M$ is known, we do not have any free parameter in this construction. It is easy to see that the number of non-zero entry is $2n + 8$. This finishes the proof. □

Now we are ready to prove Theorem 3.5.

*Proof of Theorem 3.5.* To prove our main result on polynomial generation, we first apply Lemma 3.3. Since the matrix $F \in \mathbb{R}^{n_q \times d}$ can be obtained by training, we set $F = [\xi_1, \cdots, \xi_{n_q}]^\top$ and let $\xi_i$ to be those vectors we need in Lemma 3.3 for $i = 1, \cdots, n_q$. Then we know that the inputs for the transformer encoder blocks are
$$z_i = [\xi_i \cdot x, \overbrace{0, \cdots, 0, \underbrace{1}_{(i+1)-\text{entry}}, 0, \cdots, 0}^{n}, \overbrace{0, \cdots, 0}^{q}, 1]^\top,$$
for $i = 1, \cdots, n_q$. Then we only need to apply Lemma 3.1 $q$ times with suitable adjustments of the position of non-zero entries to make sure that the product of two elements in vectors are saved in a right entry.

For the first transformer encoder block, we calculate the product of $\xi_i \cdot x$ and 1 and place $-\xi_i \cdot x$ it in the $(n_q + 2)-$th entry. Since we know that $\|\xi_i\| = 1$, if we further assume that $\|x\| < B$, then we have $|\xi_i \cdot x| \le B$. Then we only need to set $M = B$ in Lemma 3.1 and the output vectors are
$$z_i = [\xi_i \cdot x, e_i, \overbrace{-\xi_i \cdot x, \cdots, 0}^{q}, 1]^\top,$$
where $e_i$ denotes the one-hot vector of dimension $n$ with value 1 in the $i$-th position of $e_i$. For the second transformer encoder block, we calculate the product of $\xi_i \cdot x$ and $-\xi_i \cdot x$ to get $(\xi_i \cdot x)^2$ and place it in the $(n_q + 3)-$th entry. We set $M = B$ in Lemma 3.1 and the output vectors are
$$z_i = [\xi_i \cdot x, e_i, \overbrace{-\xi_i \cdot x, (\xi_i \cdot x)^2, \cdots 0}^{q}, 1]^\top,$$

Without loss of generality, we set $q$ to be odd. For the $i$-th block with $i = 3, \cdots, q$, we set $M = B^{i-1}$. Then after $q$ transformer encoder blocks, the outputs are

$$z_i = [t_i, e_i, \overbrace{-t_i, t_i^2, \cdots, -t_i^q}^{q}, 1]^\top \in \mathbb{R}^{(n+q+2) \times 1},$$

where $t_i = \xi_i \cdot x$ for $i = 1, \cdots, n_q$. Now we have different powers of $\xi_i \cdot x$ for $i = 1, \cdots, n_q$. Then we only need to set elements of $\beta$ as those $\beta_{k,s}$ we need in Lemma 3.3 and $b = Q(0)$ to generate the polynomial $Q$ we want.

Since we assume that $B$ is known, then there is no free parameter in transformer encoder blocks. The free parameters in our model all come from $F$, $\beta$ and $b$. By $n_q = \binom{d-1+q}{q}$, it is easy to see that $n_q \le d^q$. The number of free parameters in $F$ is less then $d^{q+1}$. Since for each $z_i$, we only need $q$ non-zero entries in $\beta$, the number of free parameters in $\beta$ is less then $qd^q$. So the total number of free parameters is less than $d^{q+1} + qd^q + 1$.

The number of non zero entries in this model is those in $F$, $W^K$, $W^Q$, $W^V$, $W_1$, $W_2$, $b_1$ $b_2$ in each block and in $\beta$, $b$. It can be calculated easily to know the number of non zero entries is less than $d^{q+1} + 3qd^q + 8q + 1$.

This finishes the proof. □

We first present a bound for covering number of our hypothesis space $\mathcal{H}$ (2) as stated in the lemma below. The covering number $\mathcal{N}(\eta, \mathcal{H})$ of a subset $\mathcal{H}$ of $C(\mathcal{X})$ is defined for $\eta > 0$ to be the smallest integer $l$ such that $\mathcal{H}$ is contained in the union of $l$ balls in $C(\mathcal{X})$ of radius $\eta$.

**Lemma A.1.** *For $q \in \mathbb{N}$ and $\mathcal{H}$ given in (2), with two constants $C_{q,d,\tilde{B},B}$ and $C'_{q,d,\tilde{B},B}$ depending on $Sq, d, \tilde{B}, B$, there holds*

$$\log\left\{\mathcal{N}\left(\hat{\delta}, \mathcal{H}\right)\right\} \le (d+q+1)n_q \log\left\{\frac{1}{\hat{\delta}}\right\} + C'_{q,d,\tilde{B},B} n_q \log\{n_q\},$$

*where $C_{q,d,\tilde{B},B} = 6\tilde{B}^2 q^2 dB^q$ and $C'_{q,d,\tilde{B},B} = (d+q+1)\left(\log\left(C_{q,d,\tilde{B},B}\right) + 1\right)$.*

*Proof of Lemma A.1.* For fixed $q$, if $\hat{\mathcal{C}}^q(x)$ is another function from the hypothesis space induced by $\hat{\beta}, \hat{\xi}_i, , \hat{b}$ satisfying the restrictions in (2) and

$$\left\|\beta_i - \hat{\beta}_i\right\|_\infty \le \delta, \left\|\xi_i - \hat{\xi}_i\right\|_\infty \le \delta, \left\|b - \hat{b}\right\|_\infty \le \delta,$$

then we have

$$\left\|\beta \cdot \mathbf{concat}\left(\mathcal{T}^q(x)\right) + b - \hat{\beta} \cdot \mathbf{concat}\left(\hat{\mathcal{T}}^q(x)\right) - \hat{b}\right\|_\infty$$
$$\le \left\|\beta \cdot \mathbf{concat}\left(\mathcal{T}^q(x) - \hat{\mathcal{T}}^q(x)\right)\right\|_\infty + \left\|\left(\beta - \hat{\beta}\right) \cdot \mathbf{concat}\left(\hat{\mathcal{T}}^q(x)\right)\right\|_\infty + \left\|b - \hat{b}\right\|_\infty$$
$$\le q n_q \tilde{B} d\delta BqB^{q-1} + q n_q \delta B^q + \delta$$
$$\le 3\tilde{B}q^2 dB^q n_q \delta := \hat{\delta}$$

Then, by taking an $\delta$-net for each of $\xi_i$, $\beta$ and $b$, we know that the covering number of the hypothesis space $\mathcal{H}$ with radius $\hat{\delta} \in (0, 1]$ can be bounded as

$$\mathcal{N}\left(\hat{\delta}, \mathcal{H}\right) \le \left\lceil\frac{2}{\delta}\right\rceil^{dn_q} \left\lceil\frac{2\tilde{B}}{\delta}\right\rceil^{qn_q+1}$$
$$\le \left(\frac{1}{\delta}\right)^{dn_q+qn_q+1} (2\tilde{B})^{dn_q+qn_q+1}$$
$$\le \left(\frac{1}{\hat{\delta}}\right)^{dn_q+qn_q+1} (C_{q,d,\tilde{B},B} n_q)^{dn_q+qn_q+1},$$

where $C_{q,d,\tilde{B},B} = 6\tilde{B}^2 q^2 dB^q$. Thus we have

$$\log\left\{\mathcal{N}\left(\hat{\delta},\mathcal{H}\right)\right\} \leq (d+q+1)n_q \log\left\{\frac{1}{\hat{\delta}}\right\} + C'_{q,d,\tilde{B},B} n_q \log\{n_q\},$$

where $C'_{q,d,\tilde{B},B} = (d+q+1)\left(\log\left(C_{q,d,\tilde{B},B}\right)+1\right)$.

$\square$

*Proof of Theorem 3.8.* With suitable scaling transformation, we can directly apply Lemma 4 in Schmidt-Hieber et al. (2020) or Lemma 5 in Oono & Suzuki (2019) to get

$$\mathbb{E}\left\|f_{D,\mathcal{H}} - f_\rho\right\|_{\rho_\mathcal{X}}^2 \leq (1+\epsilon)^2 \left[\inf_{f\in\mathcal{H}} \mathbb{E}[(f(x)-f_\rho(x))^2]\right] + F^2 \frac{18\log\mathcal{N}(\delta,\mathcal{H})}{N\epsilon} + 32\delta F$$

for all $\epsilon, \delta > 0$. Here $F$ is the sup-norm of functions in $\mathcal{H}$ and we can take $F = 2qn_q\tilde{B}B^q$. By applying Lemma A.1, $\epsilon = 1$, $\delta = \frac{1}{N}$ and $N \geq n_q$, we have

$$\mathbb{E}\left\|f_{D,\mathcal{H}} - f_\rho\right\|_{\rho_\mathcal{X}}^2 \leq 144C''_{q,d,\tilde{B},B} \frac{n_q^3 \log N}{N} + 64q\tilde{B}B^q \frac{n_q}{N},$$

where $C''_{q,d,\tilde{B},B} = C'_{q,d,\tilde{B},B} q^2 \tilde{B}^2 B^{2q}$. Thus we have

$$\mathbb{E}\left\|f_{D,\mathcal{H}} - f_\rho\right\|_{\rho_\mathcal{X}}^2 \leq \tilde{C}_{q,d,\tilde{B},B} \frac{n_q^3 \log N}{N},$$

with $\tilde{C}_{q,d,\tilde{B},B} = 144C''_{q,d,\tilde{B},B} + 64q\tilde{B}B^q$. This finishes the proof.

$\square$

# B EXPERIMENTAL DETAILS

In this section, we describe the additional details of our experiments.

## B.1 EXPERIMENTAL SETTING

**Data generating process** For the target function $f_{\rho,1}$, we generate 10000 i.i.d. sample $\mathbf{x}$ from a multivariate Gaussian distribution $\mathcal{N}(\mathbf{0}, \Sigma_1)$ with $\Sigma_1 = \text{diag}(100, 100)$. We randomly choose 9000 of them for training and 1000 data for testing.

For the target function $f_{\rho,2}$, we generate 50000 i.i.d. sample $\mathbf{x}$ from a multivariate Gaussian distribution $\mathcal{N}(\mathbf{0}, \Sigma_2)$ with $\Sigma_2 = \text{diag}(1, \cdots, 1) \in \mathbb{R}^{10\times10}$. We randomly choose 45000 of them for training and 5000 data for testing.

**Training Hyper-parameter** In all the experiments, we use SGD optimizer with one cycle learning rate Smith & Topin (2019), with an initial learning rate 0.0001 and maximum learning rate 0.001. For the polynomial $f_{\rho,1}$, we train three models 600 epochs with batch size 5000, and for the polynomial $f_{\rho,2}$, we train three models 2000 epochs with batch size 25000. The gradient clipping is used for all three models to avoid gradients exploding at the beginning of training.

**Model architectures** Table 5 and 6 illustrate the architecture of two types of ReLU fully connected neural networks with a comparable number of free parameters used in Section 5. The $\text{NN}_{width}$ has the same kind of linear transformation from $\mathcal{R}^d \to \mathcal{R}^{n_q}$ as our single-head self-attention transformer, while the $\text{NN}_{depth}$ has the same hidden layer $q+1$ as our single-head self-attention transformer.

## B.2 SENSITIVITY OF THE DEPTH OF OUR ATTENTION MODEL

For real applications, we do not know the order of the oracle polynomial $q$ in advance. Therefore, we give a sensitive analysis of the depth of our ATTENTION model in this subsection. Table 7

Table 5: The architecture of $NN_{width}$ and $NN_{depth}$ for the target polynomial $f_1^*$.

| LAYER | NN_WIDTH | NN_DEPTH | |
|---|---|---|---|
| 1 | LINEAR(IN=2,OUT=10) | LINEAR(IN=2,OUT=4) | |
| 2 | RELU | RELU | |
| 3 | LINEAR(IN=10,OUT=1) | LINEAR(IN=4,OUT=4) | $\Big\} \times 2$ |
| 4 | | RELU | |
| ... | | | |
| 7 | | LINEAR(IN=4,OUT=1) | |

Table 6: The architecture of $NN_{width}$ and $NN_{depth}$ for the target polynomial $f_2^*$.

| LAYER | NN_WIDTH | NN_DEPTH | |
|---|---|---|---|
| 1 | LINEAR(IN=10,OUT=4368) | LINEAR(IN=10,OUT=120) | |
| 2 | RELU | RELU | |
| 3 | LINEAR(IN=4368,OUT=1) | LINEAR(IN=120,OUT=120) | $\Big\} \times 5$ |
| 4 | | RELU | |
| ... | | | |
| 13 | | LINEAR(IN=120,OUT=1) | |

demonstrates that when the number of layers is smaller than the order of the polynomial, we cannot get the zero approximation error, while when the number of layers is larger than or equal to the underlying order, our ATTENTION model consistently gives good approximation results. This experiment demonstrates our ATTENTION model is not sensitive to the choice of the depth. For real applications, we can increase the depth of our ATTENTION model when the training error is not small enough.

Table 7: The comparison of Mean squared error for learning $f_{\rho,4} = x_1^3 + x_2^3$ by using our ATTENTION model with different number of layers.

| | ORACLE(DEPTH=3) | DEPTH=2 | DEPTH=4 | DEPTH=5 |
|---|---|---|---|---|
| $MSE_{TRAIN}$ | 0.9906 | 13.20 | 0.9986 | 0.9899 |
| $MSE_{TEST}$ | 0.0002 | 11.42 | 0.00037 | 0.00036 |

### B.3 EXPERIMENT RESULT ON THE REAL-WORLD DATA-SET

Table 8 demonstrates the effectiveness of our ATTENTION model on Household Electric Power Consumption data-set Hou. This data-set contains 2,075,260 samples gathered between 16/12/2006 and 26/11/2010. We use several properties (global reactive power, voltage, global intensity, and three substrings) to predict global active power and construct train/test sets with a ratio of 7 : 3 after removing all the null data.

### B.4 EXTENSION ON THE IMAGE CLASSIFICATION TASK

The architecture illustrated in Figure 1 is designed for regression tasks. And our experiments on synthetic data and real-world data-set verify our theoretical results. In this subsection, we aim to demonstrate the potential of our proposed model on the image classification task.

**Model Architecture**   Since our model is designed for the regression setting, we try to modify our model to fit the classification task. There are two major modifications for generalizing our model to image classification task. First, we borrow the idea of Vision Transformer Dosovitskiy et al.

Table 8: The comparison of Mean Squared Error for real-world data-set, Household electric power consumption data-set. Column ATTENTION demonstrates the MSE achieved by our architecture, while Column $NN_{depth}$ and $NN_{width}$ demonstrates the MSE achieved by the fully connected neural networks with comparable amount of parameters.

|  | ATTENTION | $NN_{depth}$ | $NN_{width}$ |
|---|---|---|---|
| $MSE_{TRAIN}$ | **0.0044** | 0.9038 | 0.0051 |
| $MSE_{TEST}$ | **0.0094** | 0.9017 | 0.0097 |

(2021) to adjust our architecture. We split an image into several non-overlapped patches and adopt a linear transformation to these patches. After the concatenation, we feed the resulting sequence of embedding to our ATTENTION model. Similar to Dosovitskiy et al. (2021), we also adopt the position encoding and extra learnable classification token. Second, like standard classification model, we use a classification head to transform the representation to $\mathcal{R}^c$, where $c$ denotes the class number.

**Experiment setting**   To demonstrate the advantage of our self-head ATTENTION model, we compare the classification results with the original Vision Transformer Dosovitskiy et al. (2021) on CIFAR-10 data-set. CIFAR-10 data-set Krizhevsky (2009) contains 32x32 color training images in 10 classes. There are 50k training images and 10k testing images.
We try different model setting for a comprehensive comparison. For VIT, we set the layer as 12, the number of patch as 4, 8, or 16, and the number of attention head as 12. For our single-head ATTENTION model, we set the layer as 2 and the number of patch as 4, 8, or 16. Moreover, we also try to adopt the hybrid architecture proposed in Section 3.1 in Krizhevsky (2009), i.e., instead of using the raw image patches, we use the feature map encoded by the CNN model.

**Training Hyper-parameter**   In all the experiments, we use SGD optimizer with one cycle learning rate Smith & Topin (2019), with an initial learning rate 0.001 and maximum learning rate 0.01. We train each model for 200 epochs with batch size 512.

**Experimental results**   Table 9 demonstrates the effectiveness of our single-head ATTENTION model on CIFAR-10 data-set. Our model consistently gives better results on different patch size even though the weights in our ATTENTION block do not need to train. These experimental results further confirms the potential of our proposed ATTENTION model.

Table 9: The comparison of testing accuracy on CIFAR-10 data-set. VIT denotes the vision transformer and ATTENTION denotes our model.

|  | TEST ACC(%) |
|---|---|
| VIT LAYER=8 PATCH=16 | 65.36 |
| VIT LAYER=12 PATCH=8 | **74.92** |
| VIT LAYER=12 PATCH=4 | **81.08** |
| VIT HYBRID | 81.26 |
| ATTENTION PATCH=16 | **73.34** |
| ATTENTION PATCH=8 | 73.02 |
| ATTENTION PATCH=4 | 71.44 |
| ATTENTION HYBRID | **85.43** |

**Further discussion**   There are some interesting studies of our proposed single-head ATTENTION model.

1. Why the performance of the Vision Transformer is much lower than that reported in Krizhevsky (2009)?

2. Whether a smaller patch size can improve the performance of the results of our ATTENTION model?

3. Whether the hybrid architecture can further improve performances of the model?

We give detailed discussion of the above questions.

1. The accuracy of VIT is different from the Table 2 in Krizhevsky (2009) because we train the model on CIFAR-10 from sketch, while the author in Krizhevsky (2009) pretrain the VIT on huge vison data-set, i.e., ImageNet, ImageNet-21k, and JFT-300M, and finetune it on CIFAR-10. The performance of VIT drops dramatically when the number of sample becomes smaller. Our ATTENTION model can achieve better results than VIT without training the attention weights and with much smaller attention heads might give some inspirations for further design of the Transformer model on the small data regime.

2. The larger patch size gives a better performance for our ATTENTION model, which is different from the case of VIT. This phenomena matches the design of the data-preprocessing process in our ATTENTION model well since it encourages us to use linear transformation on the full input data to capture some global information.

3. Using a hybrid architecture can improve the performance of our ATTENTION model dramatically (+12.41%). This illustrates the superiority of the convolution architecture to capture local information of image data than linear transformation, which gives some insights to further improve the ATTENTION architecture.

