# OpenReview forum: "Attention Enables Zero Approximation Error"
_ICLR.cc/2023/Conference — Submitted to ICLR 2023_

### Official Review · Reviewer_HdBP · 2022-10-24

**Confidence:** 3
**Correctness:** 3
**Technical Novelty And Significance:** 4
**Empirical Novelty And Significance:** 2
**Recommendation:** 5

**Clarity, Quality, Novelty And Reproducibility:**


Clarity: fair, in general it can be understood with some effort.

Quality: the theory analysis is good; the experiments are weak.

Novelty: very novel.

Reproducibility: good. It will be easy to implement such a network.


**Strength And Weaknesses:**

Strength:

- The presented theoretical analysis framework is novel.
- The conclusion that a Transformer can universally approximate a polynomial function is strong.

Weakness:

- The study assumes that the input has a fixed size, i.e., $x \in R^d$ where d is a constant. This is not true when a Transformer is used for sequence modeling.
- The preprocessing only uses matrix F to embed the x, not considering positional embedding, which is important for Transformer.
- The main experiment is only on synthetic data.


**Summary Of The Paper:**

This paper presents a novel theoretical analysis of the Transformer networks, showing that with a special preprocessing, Transformer can universally approximate any polynomial function. The theoretical analysis is based on a simplified transformer which only has a single attention head and only handle fixed input size. The authors also show that the proposed method can avoid the approximation error and sample error trade-off. Finally, the authors provided experimental results to verify their analysis.

**Summary Of The Review:**

Overall, it is a novel perspective to understand a Transformer and has a lot of potential. My main concern is about the fixed length of input, because (1) Transformer is a powerful network in sequence modeling; (2) previous works, e.g., Yun et al. (2019), already can analyze the sequential input.

---

> ### Author Response · Authors · 2022-11-16
> **For Reviewer HdBP**
>
> We thank the reviewer for the encouraging comments and suggestions. We appreciate the time you spent on the paper. Below we address the concerns and comments that you have provided.
>
> **Q**: *The study assumes that the input has a fixed size, i.e., $x \in R^d$ where d is a constant. This is not true when a Transformer is used for sequence modeling.*
>
> **A**: First, in our paper, it is easy to change the output dimension from one-dimensional case to multi-dimensional case by increasing the number of heads or by setting $\beta$ into a matrix which both fit in the case of sequence-to-sequence model. For example, our single head model maps vectors in $\mathbb{R}^d$ to $\mathbb{R}$. Thus we can map a vector in $\mathbb{R}^d$ to $\mathbb{R}^d$ by setting the heads of attention to $d$. Besides, in order to present a quantifiable structure of the corresponding model, it is common to consider the dimension $d$ as a constant. We can set $d$ large enough to handle inputs of different lengths.
>
> **Q**: *The pre-processing only uses matrix F to embed the x, not considering positional embedding, which is important for Transformer.*
>
> **A**: From a high-level idea, we introduce the matrix F to focus on global information rather than local information. So considering from this perspective, the positional embedding is not suitable in our framework. And, in Yun et al. (2019), the positional embedding is applied to remove the permutation equivariant condition of target functions for the universal approximation ability of the Attention/Transformer. There is no such condition in our paper, so we have not considered introducing the positional embedding for the time being. This also shows the superiority of our proposed model.
>
> **Q**: *The main experiment is only on synthetic data.*
>
> **A**: As mentioned in the contribution part in Section 1, we conduct the experiment on real-world data-set for regression tasks. The performances of our proposed model is much better than traditional fully connected neural networks with a comparable number of free parameters. We also apply our model to the image classification task and achieve better performance than Vision Transformer on the CIFAR-10 data set with suitable adaptations. The detailed results can be found in the last paragraph in Section 5 and in Appendix B.3 and B.4.

---

### Official Review · Reviewer_8KLD · 2022-10-24

**Confidence:** 5
**Correctness:** 3
**Technical Novelty And Significance:** 3
**Empirical Novelty And Significance:** 3
**Recommendation:** 5

**Clarity, Quality, Novelty And Reproducibility:**

The paper is clear, while some of the statements are a little overclaimed. Overall, the results are interesting but natural because, with the inner product structure of attention, it is easy to construct the basic components to form a polynomial.

**Strength And Weaknesses:**

Pros:
- The construction is clear and interesting.
Cons:
-  Most of the claims in the paper are just simple consequences of the exact construction. For example, "the encoder blocks do not need to be trained" and "avoiding the classical trade-off between approximation error and sample error" are just because the construction gives exact polynomials. However, the authors intentionally highlighted these as new contributions, making the paper a little overclaimed.
- The comparison of the transformer and fully connected network in this paper is not fair because it is obvious that the transformer can compute functions over different tokens, while the only way for the fully connected network considered in this paper to mix different tokens is by using the linear combination in the last layer. Given the discussion in the paper is limited to fixed-length input, a better baseline would be a fully connected network over different tokens.

**Summary Of The Paper:**

This paper proves that transformers with single-head attention and a specific embedding scheme can express any polynomial by constructing the weights required for forming a polynomial.

**Summary Of The Review:**

In summary, the paper shows a clear construction of how to express polynomials with transformers, but the significance is borderline.

---

> ### Author Response · Authors · 2022-11-16
> **For Reviewer 8KLD**
>
> We thank the reviewer for the encouraging comments and suggestions. We appreciate the time you spent on the paper. Below we address the concerns and comments that you have provided.
>
> **Q**: *Most of the claims in the paper are just simple consequences of the exact construction. For example, "the encoder blocks do not need to be trained" and "avoiding the classical trade-off between approximation error and sample error" are just because the construction gives exact polynomials. However, the authors intentionally highlighted these as new contributions, making the paper a little overclaimed.*
>
> **A**: Thank you for your comments. The main contribution of our work is the zero approximation error part. Therefor, it is possible to avoid the classical trade-off between approximation error and sample error. Since in our construction we only use constants, we say that encoder blocks do not need to be trained. Of course, this part of the constant can also be achieved through training. However, this does not affect the fact that we can construct the target polynomial with finite number of free parameters.
>
> **Q**: *The comparison of the Transformer and fully connected network in this paper is not fair because it is obvious that the Transformer can compute functions over different tokens, while the only way for the fully connected network considered in this paper to mix different tokens is by using the linear combination in the last layer. Given the discussion in the paper is limited to fixed-length input, a better baseline would be a fully connected network over different tokens.*
>
> **A**:For the fully connected network, it mix different tokens from the first layer rather than in the last layer mentioned by the reviewer. Therefore, we believe it is a fair comparison with comparable amount of free parameters,.

---

### Official Review · Reviewer_sof3 · 2022-10-27

**Confidence:** 4
**Correctness:** 2
**Technical Novelty And Significance:** 2
**Empirical Novelty And Significance:** 1
**Recommendation:** 3

**Clarity, Quality, Novelty And Reproducibility:**

**Clarity**: the mathematical results are well detailed. The overall language can be improved. A section on related work on theoretical analysis of transformers (with more papers than the 3 in section 4) should be added.

**Quality**: As I explained above, there is a problem with the core results of the paper. Their actual content and limited impact contrasts with what the authors claim they achieve.

**Novelty/Originality**: On the theoretical side, the overall approach of showing that fixed sets of weights in popular architectures turn them into classic functions seems novel and original. It would however be more convincing if the authors did not change the softmax layer (or changed it to a hard attention). On the experimental side the paper shows that standard machine learning models can outperform neural networks on some small datasets. This seems hardly novel to me.

**Reproducibility**: Experimental hyperparameters are well detailed in appendix, and the authors provide their code. One minor issue in section 5.1 is that sampling function of input $x$ should be mentioned. It's essential to understand the results, especially how shallow networks behave on that data, and should not be in appendix.

**Strength And Weaknesses:**

**Strengths**: The paper's claims are well detailed. The authors provide a complete formalism, and despite technicality the results and proofs are relatively easy to follow. Theoretical results seem correct and I have no reason to doubt the empirical ones.

**Weaknesses**: Although I don't technically doubt them, I think that the core findings of the paper are weaker and less novel than the authors claim them to be, and could be considered misleading. I have two major concerns.

First, the authors mention that the softmax in the transformer encoder block is replaced with "one hot maximization". This is all but an innocuous change. Softmax outputs a probability distribution which at the limit becomes hard attention (the largest value becomes 1 and the others 0). But one hot maximization keeps the largest value in the attention scores. After multiplying those scores with the values, the output of the self-attention layer is quadratic. That makes it easy to use to model polynomials, but it hardly says anything about standard self-attention. Therefore the claim in section 4 that the authors "guarantee the integrity of transformer encoder blocks" is misleading.

Second, the authors fix the transformer weights in all experiments in order to model polynomials. The output is always of the form $t,t^2,...,t^n$ with $t$ the input. The only trained weights in an initial linear layer $F$, and most importantly weighting coefficients of the outputs. In other words, the model they train is just a linear model on the outputs of $F$ in a polynomial feature space. That such a model is good at modeling polynomials is obvious, and that it performs better than a large neural network on small datasets is not particularly surprising either.

I would argue that the paper is not really about transformers or self attention. They show that a model which isn't really a transformer but contains inner products can with specific weights be set to a polynomial feature extractor - though not a very efficient one. There are certainly interesting aspects of that result but they are not the ones the paper focuses on.

**Summary Of The Paper:**

This paper analyses the potential for attention-based networks to model polynomial functions. The approach consists in taking a variation of the single-head transformer, and show that with proper input preprocessing and a specific set of fixed weights it can be set to any polynomial. The authors back their claims with mathematical proofs, and confirm them with experimental results on synthetic data. They also show that such a model can perform better than a Visual Transformer on the CIFAR10 dataset.

**Summary Of The Review:**

Although taking an interesting approach, the paper does not convey compelling results in a convincing manner. The vocabulary used (attention, transformer, etc) is misleading with respect to the actual results, which say very little about transformers and (standard) attention. I tend to consider this a correctness issue: even though the theorems and lemmas are correct, the conclusions drawn from them are not.

I do not think that this paper in its current form should be accepted.

---

> ### Author Response · Authors · 2022-11-16
> **For Reviewer sof3**
>
> We thank the reviewer for the comments. We appreciate the time you spent on the paper. Below we address the concerns and comments that you have provided.
>
> **Q**: *The authors mention that the softmax in the transformer encoder block is replaced with ''one hot maximization".*
>
> **A**: Thank you very much for your comment. As we pointed out in the paper, we have made some modifications to the Transformer model, which is not completely in line with the actual structure in practice. Our ultimate goal, of course, is to give a reasonable explanation of why the structure in practical application can be successful. The research idea we adopted is to simplify the model first, and then gradually restore it to the actual appearance. In such a process, we found that if we replace soft-max to one-hot-maximum, we can express any polynomial by this model. In the work Yun et al. (2019), as we pointed out in the discussion, the self-attention layers and feed-forward layers are set in an alternate manner. This is not a way to analyze the encoder block as a whole, so we say that we guarantee the integrity of transformer encoder blocks.
>
> **Q**: *The authors fix the transformer weights in all experiments in order to model polynomials.*
>
> **A**: Thank you very much for your comment. Since in our construction we only use constants, we say that encoder blocks do not need to be trained. Of course, this part of the constant can also be achieved through training. However, this does not affect the fact that we can construct the target polynomial with finite number of free parameters.

---

> > ### Comment · Reviewer_sof3 · 2022-11-16
> > **response to authors**
> >
> > *Of course, this part of the constant can also be achieved through training*
> > This does not seem certain to me. Although you show that some fixed set of weights models a polynomial with no error, it does not mean that training those weights would lead to the same solution. Most optimizers would probably prefer a different solution with some approximation error, but less sparse matrices.
> >
> > This however is besides my main concern, which is that your your model, stripped of the transformer/attention vocabulary, just acts like a polynomial feature extractor. On top of that you apply a maximum entropy classifier. Granted, that leads to a model with few finite parameters, but that's just standard Machine Learning. You show that:
> > * polynomial features of degree $n$ can be expressed by with $\mathcal{O}(n)$ matrix multiplications and dot products (which isn't so impressive when $x\rightarrow x^n$ is a classic $\mathcal{O}(log (n))$ algorithm)
> > * MaxEnt in polynomial space is still competitive if the dataset is synthetic or small
> >
> >  I don't think that these are sufficient contributions.

---

### Official Review · Reviewer_H8Qq · 2022-10-31

**Confidence:** 4
**Correctness:** 3
**Technical Novelty And Significance:** 3
**Empirical Novelty And Significance:** 3
**Recommendation:** 6

**Clarity, Quality, Novelty And Reproducibility:**

Clarity: Fair
Quality: Fair
Novelty: Fair
Reproducibility: Fair

**Strength And Weaknesses:**

Strength: This paper provides a theoretical view towards the Transformer block and the theoretical results are interesting. Even though the experiment data is synthetic, the experiments can back up their experiment results.

Weaknesses:
(i) In Section 4, the authors discuss the difference between their paper and the previous paper. One difference is that prior papers focus on seq-to-seq tasks and they focus on regression tasks. However, I think seq-to-seq tasks are more universal than the latter one. Thus, is the conclusion weaker than the previous one?
(ii) The authors introduce the single-head Transformer but ignore the effectiveness of multi-head. Can you explain the effect of the number of heads in your theoretical results? As all we know, the multi-head mechanism can improve performance.


**Summary Of The Paper:**

This paper introduces a single-head self-attention transformer model and shows that any polynomial function could be generated exactly by an output function of such a model with the number of transformer encoder blocks equal to the degree of the polynomial. Furthermore, these transformer encoder blocks in this model do not need to be trained. Various experiments and ablation studies to verify our theoretical results demonstrate the effectiveness of their theoretical results.

**Summary Of The Review:**

This paper provides theoretical results for single-head Transformer and the results are interesting. They also perform various experiments to support their conclusions.

---

> ### Author Response · Authors · 2022-11-16
> **For Reviewer H8Qq**
>
> We thank the reviewer for the encouraging comments and suggestions. We appreciate the time you spent on the paper. Below we address the concerns and comments that you have provided.
>
> **Q**: *One difference is that prior papers focus on seq-to-seq tasks and they focus on regression tasks. However, I think seq-to-seq tasks are more universal than the latter one. Thus, is the conclusion weaker than the previous one?*
>
> **A**: Our analysis can be easily extended to sequence-to-sequence tasks by increasing the number of heads or by setting $\beta$ into a matrix. For example, our single head model maps vectors in $\mathbb{R}^d$ to $\mathbb{R}$. Thus we can map a vector in $\mathbb{R}^d$ to $\mathbb{R}^d$ by setting the heads of attention to $d$. If we further assume that each output dimension is a polynomial of the input, zero approximation error and the universal property can also be achieved.
>
> **Q**: *The authors introduce the single-head Transformer but ignore the effectiveness of multi-head. Can you explain the effect of the number of heads in your theoretical results?*
>
> **A**: As we discussed in the above question, increasing the number of heads can generalize our model to sequence-to-sequence task. Besides, it is obvious that we can also use multi-head model to learn some more complex functions. For example, a function $g(f_1(x),\cdots,f_n(x))$ with hierarchical structure where each $f_i$ is a polynomial of $x$ can be learned by our proposed model with $n$-head easily.

---

### Official Review · Reviewer_4Zo8 · 2022-11-01

**Confidence:** 4
**Correctness:** 3
**Technical Novelty And Significance:** 3
**Empirical Novelty And Significance:** Not applicable
**Recommendation:** 5

**Clarity, Quality, Novelty And Reproducibility:**

The author included many details in the supplementary material, which is convincing. And the author compared this work with Yun et al. (2019) heavily, which is very necessary. However, I am not convinced by the comparison result.

In Yun et al. (2019), they explored the representation power of the transformer by showing that it is a universal approximation of sequence-to-sequence functions with less number of required parameters compared with the residual network (Section 4.4 in Yun et al. (2019)).

In addition, this paper limits the target function as polynomial and the task as regression and classification, but Yun et al. (2019) do not have such requirements.

Further, it is true that Yun et al. (2019) is the approximation, but their 'error' between continuous functions and piece-wise functions, transformers and modified transformers, and piece-wise functions and modified transformers are infinitesimal. I could not find the advantage of using polynomials directly compared with the universal approximation theory Yun et al. (2019) adapted.

Finally, yes Yun et al. (2019) is not focused on the dot product (or the attention matrix), but many following works covered this direction. Such as Yun et al. (2020), Zaheer et al. (2020), and Shi et al. (2021).

**Strength And Weaknesses:**

Strength:
1. The number of required transformer blocks is relatively small.
2. The transformer encoder is not needed to be trained, with the cost of introducing free parameters F, \beta, and b.
3. The modification is relatively simple compared with the original scale single-head self-attention layer.

Weakness:
1. Only discussed the upper bound of free parameters, which is very large.
2. In my understanding, most of the analysis is based on the assumption that the target function is polynomial.
3. Limits its application to regression and classification.

**Summary Of The Paper:**

This paper shows the single-head self-attention transformer with a fixed number of transformer encoder blocks and free parameters could generate the desired polynomial of input directly without approximation. Further, the author shows that with the increasing number of free parameters and polynomial of degree q, the single-head self-attention transformer block is universal.

**Summary Of The Review:**

As above, my main concern is:
1. This theory included two strong assumptions, which limits its application value.
2. Compared with Yun et al. (2019), the benefit is not obvious to me.

---

> ### Author Response · Authors · 2022-11-16
> **For Reviewer 4Zo8**
>
> We thank the reviewer for the encouraging comments and suggestions. We appreciate the time you spent on the paper. Below we address the concerns and comments that you have provided.
>
> **Q**:*Only discussed the upper bound of free parameters, which is very large.*
>
> **A**: As far as we are concerned, no existing work shows that a network architecture can achieve zero approximation error for polynomials with finite parameters. The universal approximation theory can only guarantee $\epsilon$-approximation error with either arbitrary width and bounded depth neural network or bounded width and arbitrary depth network. In this sense, showing that an architecture can achieve zero approximation error with finite number of parameters is a significant contribution itself.
>
> **Q**:*In my understanding, most of the analysis is based on the assumption that the target function is polynomial.*
>
> **A**: Yes. Polynomial functions are the most powerful tool for analyzing various function classes, and our analysis can be easily extended to other tasks. In most of the existing deep learning theoretical papers, it is often necessary to approximate polynomial functions by the model and then approximate the objective function class through the polynomials. As we pointed out in the paper, our model can generate arbitrary polynomial functions, which is fundamentally different from models like a fully connected network. Due to the use of piece-wise linear activation functions, traditional neural network models cannot perfectly generate arbitrary polynomial functions. This makes our proposed model stand out.
>
> **Q**:*Limits its application to regression and classification.*
>
> **A**: Our analysis can be easily extended to sequence-to-sequence tasks by increasing the number of heads or by setting $\beta$ into a matrix. For example, our single head model maps vectors in $\mathbb{R}^d$ to $\mathbb{R}$. Thus we can map a vector in $\mathbb{R}^d$ to $\mathbb{R}^d$ by setting the heads of attention to $d$. If we further assume that each output dimension is a polynomial of the input, zero approximation error and the universal property can also be achieved.
>
> **Q**:*This theory included two strong assumptions, which limits its application value.*
>
> **A**: For the polynomial assumption, we have illustrated the importance of this function class in above answer. And we also present a result of universal approximation of our proposed model. For the type of tasks, we can also apply our model to sequence-to-sequence task as discussed above. Obviously, zero approximation error and the universal approximation property can be maintained.
>
> **Q**:*Compared with Yun et al. (2019), the benefit is not obvious to me.*
>
> **A**: Thank you for your comments. We have two major improvements compared with Yun et al. (2019). First, in their construction, the architecture of the Transformer is not fully preserved. As they stated in their paper, self-attention and feed-forward layers are considered in an alternate manner. This does not guarantee the integrity of the structure, while our theorem keeps the original architecture of the Transformer. Second, our main contribution is that our proposed model can achieve zero approximation error for polynomials. If the target is a polynomial, which is also continuous, our result is obviously better considering approximation error or excess risk.

---

### Decision · Program_Chairs · 2023-01-20

**Decision:**

Reject

**Justification For Why Not Higher Score:**

N/A

**Justification For Why Not Lower Score:**

N/A

**Metareview: Summary, Strengths And Weaknesses:**

The paper studied theoretical properties of attention-based models for representing polynomials. It does not study the exact popular form of (self-)attention but another construction, which limits the scope of applicability of the findings. The bound on free parameters is not convincingly useful. Albeit interesting, the current form of the paper is not suitable for ICLR.

**Summary Of Ac-Reviewer Meeting:**

N/A